# Porogenic Solvents in Molecularly Imprinted Polymer Synthesis: A Comprehensive Review of Current Practices and Emerging Trends

**DOI:** 10.3390/polym17081057

**Published:** 2025-04-14

**Authors:** Gil van Wissen, Joseph W. Lowdon, Thomas J. Cleij, Kasper Eersels, Bart van Grinsven

**Affiliations:** Sensor Engineering Department, Faculty of Science and Engineering, Maastricht University, P.O. Box 616, 6200 MD Maastricht, The Netherlands

**Keywords:** molecularly imprinted polymer, porogen, polymerization methods, deep-eutectic solvents, ionic liquids, emerging porogens

## Abstract

The versatility of molecularly imprinted polymers (MIPs) has led to their integration into applications like biosensing, separation, environmental monitoring, and drug delivery technologies. This diversity of applications has resulted in a plethora of synthesis approaches to precisely tailor the materials’ properties to the specific demands. A critical, yet often overlooked, factor in MIP synthesis is the choice of porogen. Porogens play a pivotal role in defining the morphology, surface properties, swelling behavior, and binding efficiencies of the resulting MIPs. While aprotic solvents have traditionally been the standard in molecular imprinting, recent developments have expanded the variety of employed porogens accompanied by notable improvements in MIP performance. Therefore, this review aims to highlight both traditional and emerging types of porogens used in molecular imprinting, their influence on polymer properties and sorption performance, and their application across various sensing and extraction applications.

## 1. Introduction

Molecularly imprinted polymers (MIPs) represent a sophisticated class of materials designed with precision at the molecular level. These polymers are crafted to mimic the specific recognition properties of natural antibodies and receptors, albeit with greater stability and durability [1]. Unlike their biological counterparts, MIPs are synthetic and engineered through a process called molecular imprinting. This concept involves creating cavities or “imprints” within the polymer matrix that match the shape, size, and spatial distribution of functional groups of a target molecule [2]. This process begins by selecting a template molecule, around which the polymer network assembles into a mold. Once the polymerization is complete, the template molecule is removed, leaving behind complementary cavities that hold the likeness of the template’s structure, as shown in Figure 1 [3]. This imprinting mechanism imparts MIPs with the ability to selectively bind the target molecule over others of similar size and shape [4]. Therefore, their selectivity combined with high chemical and physical stability has led to their integration in numerous application fields like solid-phase extraction, biosensing, diagnostics, environmental and food safety monitoring, and catalysis [5,6,7,8,9,10].

Broadly speaking, most currently accepted approaches to MIP formulation rely on five constituent components consisting of a functional monomer, functional crosslinker, template species, thermal/photo initiator, and porogenic solvent [1]. Each of the defined components has a distinct role in the formation of the synthetic receptor, with a substantial amount of focus tending to be placed on the selection of functional monomer and crosslinker. These two building blocks are carefully selected to yield strong interactions with the template species in the form of complimentary hydrogen bonding, electrostatic (ionic), π-stacking, and Van der Waals interactions. The functional groups present in the monomer yield more meaningful interaction with the template species, whereas the main purpose of the crosslinker is to induce rigidity and to stabilize the polymeric network [11]. The ratio of these components in respect to the template species is a key aspect of rationale MIP design, with their methodical iteration facilitating the tailoring of a synthetic receptor towards a set of desired binding parameters [12].

Generally, radical polymerization is the method of choice when producing MIPs, with the use of the selected thermal or photo-initiators yielding a polymer by initiating the successive addition of free-radical building blocks, in this case functional monomer and crosslinker, thereby growing the polymeric network [13]. The most prevalent initiator of choice for MIP enthusiasts is azobisisobutyronitrile (AIBN), which can undergo thermal/photolytic cleavage forming two alkyl radicals and nitrogen gas. In addition to jump-starting the radical polymerization process, the produced nitrogen gas further impacts the structure of the MIP by causing the formation of a porous network or foam, with AIBN being utilized industrially as a blowing agent [14]. This, in turn, increases the available surface of the MIP, affecting the binding capacity of the material alongside altering the bulk diffusion pathway. Many other examples of initiators are well documented throughout the literature, though they all serve the same purpose of beginning the polymerization process. Conversely, MIPs can be formed without the use of initiator species, with methodologies such as electropolymerization utilizing the redox potentials of monomeric species and applied current to form polymers, and sol–gel processes undergoing hydrolysis and condensation to form complex networks [15,16].

The main thread that connects the use of these components, in the vast majority of cases, is the parallel application of porogenic solvent. The mode of action of a porogen refers to the compounds’ role in generating pores or voids within a material with given pore sizes and volumes, with solvent-free systems producing non-porous materials. Additionally, the porogen is a solvent that ensures that all components within the pre-polymerization mixture are solvated and can interact to their full extent, which is sometimes also challenging depending on the solubility of the target species [11,17]. This said, the porogenic solvent tends to be the most overlooked constituent of MIP formulation, with considerations focusing on the impact the solvent selection has on the selected method of polymerization rather than on the physical or chemical characteristics of the generated polymer. Tailoring of the utilized porogenic solvent therefore plays a major role on morphology, determining if the yielded polymer will be microporous, mesoporous, or macroporous, with higher polarity solvents producing denser polymer networks and apolar solvents producing more porous networks [18]. Additionally, pre-polymerization complex formation is another factor to consider, with the use of some porogens interfering with key binding interactions between template, monomer, and crosslinker. The ultimate goal is to provide enough interaction with the solvent to dissolve all the polymerization components, but not too much to interfere with the polymer–template interactions. Other critical aspects to consider with the selection of porogens are the binding site accessibility, template removal, swelling behavior, thermal properties, and mechanical properties of the resultant polymer, which define the specificity, selectivity, and binding capacity of the material. It is therefore imperative that the synthetic conditions reflect that of the desired applications’ binding environment, otherwise the specificity of the receptors can be detrimentally affected.

To this date, two reviews about porogens in material fabrication and polymer monolith fabrication, respectively, have been published [19,20]. Therefore, a review on the use of porogens in the synthesis of non-monolithic MIPs and MIPs in the field of sensing is missing. This review endeavors to investigate these concepts further, scrutinizing the different situations and classifications of porogenic solvents, building a framework for the different conditions and the impact specific porogenic selections have on the synthesis of MIPs. Specifically, we will highlight the different properties of porogens and discuss the use of aprotic solvents, protic solvents, and combinations of the two, highlighting their unique properties and implications for MIP synthesis. Beyond these traditional solvents, the applications of deep eutectic solvents (DESs) and ionic liquids (ILs) as innovative porogenic agents will be evaluated, with recent advances in the field, such as the use of supercritical fluids and cryogels, offering more radical approaches to MIP production. Furthermore, we will consider emerging techniques of mechanochemistry and solventless systems, removing the need for porogenic solvents.

## 2. Physical Properties of Porogenic Solvents

The first essential step is defining the physical properties of porogens to understand their influences on the MIP synthesis. The first essential property is the boiling point (bp) of the solvent, as it determines the temperature range in which a synthesis can be conducted. For example, the previously mentioned radical starter AIBN decomposes at a temperature of 60 °C to initiate the polymerization [21]. The main distinction of porogens in molecular imprinting is the proticity of a solvent, designating its ability to act as a hydrogen bond donor [22]. Traditionally, it is claimed that aprotic solvents perform superiorly as porogens as they do not interfere with the monomer–template complex compared to protic solvents, which could donate their hydrogen bond to any given polymerization component [23]. Furthermore, the dielectric constant or static permittivity ε is an important metric to consider when developing MIPs. In general, it is a measure of the solvent’s ability to stabilize ions in solution. A high entails ε a high degree of dissociation of electrolytes into ions, while a low ε indicates low ionic dissociation [24]. This is crucial when considering the desired template–monomer interactions, with acid–base and ionic interactions being favored in high-ε solvents and hydrogen bonding being preferred in low-ε solvents. Additional solvent characterization is obtained by considering the Hansen solubility parameter (*δ_t_*), which gives information about the potential solubility of the polymerization components in a given solvent [25]. The total Hansen parameter is divided into three distinct parameters, which describe the possible intermolecular forces based on dispersion (*δ_d_*), polar (*δ_p_*), and hydrogen-bonding forces (*δ_H_*) based on the following Equation (1) [26].(1)$$\delta_{t}=\sqrt{{\delta_{d}}^{2}+{\delta_{p}}^{2}+{\delta_{H}}^{2}}$$

Dispersion forces mainly occur between non-polar symmetrical molecules, polar forces occur between molecules of opposing polarities, and hydrogen bonding between hydrogen bond donors (HBDs) and acceptors (HBAs). The *δ_H_* describes other electron exchange forms additionally to traditional hydrogen bonding. These factors accurately describe the rule of thumb “like dissolves like” and is another potent factor to consider when designing MIPs. In contrast to Hansen parameters, which are often theoretically determined, solvatochromism is a method to empirically determine a solvent’s polarity [27]. To this purpose, a dye with a large change in its permanent dipole moment upon excitation is dissolved in a given solvent. The polarity of the solvent shifts the absorption wavelength (λ_max_) of the dye due to polar solvent–dye interactions, and this shift can be quantified. The most commonly used dye is N-phenolate betaine dye as a probe, with the polarity *E_T_*(30) [kcal/mol] being determined by the following Equation (2).(2)$$E_{T}\left(30\right)=28,591/\lambda_{max}$$

This straightforward approach allows for the determination of polarities of solvent mixtures that can potentially be applied in MIP synthesis. The bp, ε, *δ_d_*, *δ_p_, δ_H_,* and *E_T_*(30) of different protic, aprotic, and emerging solvents are summarized in Table 1 [27,28,29,30,31,32,33,34,35,36,37,38,39,40].

In the following sections, we will investigate how researchers have leveraged these solvents’ properties to develop efficient MIPs for a large diversity of analytes.

## 3. Aprotic Porogenic Solvents

An aprotic solvent is characterized by its inability to engage in hydrogen bonding because it lacks hydrogen atoms bonded to electronegative atoms like oxygen or nitrogen [41]. This property makes aprotic solvents, such as dimethyl sulfoxide (DMSO), dimethylformamide (DMF), acetone, and acetonitrile (ACN), particularly valuable in various chemical processes due to their stability and unique solvating abilities [42]. An easily apparent advantage of aprotic solvents when looking at Table 1 is the vast choice of possible porogens that are available. This allows for precise control over desired template–monomer interactions, as dielectric constants and polarities vary greatly between the solvents. The generally low *δ_H_* of the aprotic solvents indicates that the porogen will not perturb hydrogen bonding interactions between the template and monomer. For polar and ionic interactions solvents like ACN, DMSO and DMF offer great potential, while apolar and ion pairing interactions dominate in solvents like chloroform or THF. This potential is further potentiated by the ease of mixing these organic solvents to further tune their properties. Additionally, these variable properties allow for the solvation of a plethora of possible analytes. In the specific case of MIPs, the use of aprotic porogenic solvents allows for precise control over the polymerization process, enhancing the creation of imprinted cavities within the polymer matrix. These solvents help strengthen the interactions between monomers and the target molecule, rather than unintended solvent–polymer interactions. The resulting porous structure is tailored to provide optimal binding sites for the target molecule, improving the efficiency and selectivity of the MIP for its intended application.

This said, the vast majority of research conducted using aprotic solvents tends to be limited by the associated methods of polymerization, with the use of free-radical monolithic bulk and precipitation polymerization being the most common (Table 2). This narrow band of approaches is a reflection of the limitations associated with systems composed of aprotic solvents, with more sophisticated (e.g., emulsion polymerization) methodologies requiring more complex mixtures of solvents with opposing polarities [43].

While aprotic porogens enable the creation of high-affinity receptors and solvents like DMSO facilitate the solvation of nearly any template compound, a key challenge remains: the resulting receptor may only perform optimally in the porogenic solvent rather than in the analysis solvent. With this in mind, Meier et al. offer the perfect example of this with their application of aprotic porogenic solvents, ACN and chloroform in this case, for the synthesis of 4-nitrophenol MIPs [58]. In that study, both the computational and experimental aspects of the solvents in the system were studied, highlighting how the porogen plays a decisive role in determining not only the physical properties (porosity, surface area, and swelling behavior) of the MIP but also the stability of the pre-polymerization complex and thus the resulting recognitions capabilities of the material. The strongest take-away from this line of research however is the impact on porogen vs. analysis solvent (mobile phase). The resulting MIPs demonstrated similar selectivity’s (S_MIP1(ACN)_ = 1.45, S_MIP2(CHCl3)_ = 1.43) when applying the same mobile phase as synthesized porogen during the solid-phase extraction studies, yet when switching to opposing mobile phases, MIP1 lost its selectivity (S_MIP1(ACN)_ = 1.04) and MIP2 retained a decreased ability (S_MIP2(CHCl3)_ = 1.15), therefore confirming the strong dependence of the recognition properties of the MIP on the respective solvent. This study subsequently highlights one of the major challenges associated with producing MIPs in aprotic solvents opposed to solvent conditions that more closely emulate the desired environment for analysis. Producing imprints in this manner yields receptors with high affinity in the porogenic conditions, but this does not necessarily translate when utilizing other solvents for analysis. This said, many examples do exist where MIPs do perform extremely well outside their porogenic environment, meaning that other factors should be considered. As a further example, Esfandyari-Manesh et al. synthesized MIP nanospheres by precipitation polymerization for the extraction of carbamazepine (CBZ), employing 4 different porogens, namely ACN, chloroform, ACN–chloroform (1:1), and toluene [18]. The differences in morphology, surface area, and rebinding performance were investigated and can be seen in Figure 2. A huge difference in specific surface area was identified, with ACN-based MIPS having a specific surface area of 242 m^2^g^−1^ compared to 6 m^2^g^−1^ for toluene-based MIPs. All porogens resulted in mesoporous polymers, with the ACN-based MIPs showing the smallest pore diameter. This difference was also visible in the rebinding performance, as the ACN-based MIPS had the highest imprinting factor (IF) of 1.75. This research highlights that the porogen has crucial impact not only on the template–monomer interaction but also on the resulting polymer morphology, with a high specific surface area being desirable. Smaller particles have a higher surface-to-volume ratio and can adsorb more analyte, which is crucial for efficient MIPs.

Further investigation of porogens’ effect on the resulting polymer morphology was conducted by Santora et al. by measuring the surface area and pore size distribution of non-imprinted polymers (NIPs) synthesized in different solvents [59]. In general, all polymers showed a heterogeneous pore size distribution in the micro- and mesopore regime. The main takeaway from their study is the complementarity between polymer and porogen, as more apolar divinylbenzene polymers synthesized in apolar hexane have high surface areas, while methanol results in a very low surface area. Conversely, more polar EGDMA polymers synthesized in MeOH have high surface areas, while hexane results in low surface areas. Additionally, Pardeshi et al. synthesized gallic acid-imprinted polymers in ACN, THF, and acetone, with the resulting surface areas decreasing in the same order [60]. The smallest pore volume and diameter was observed for THF-based MIPs, which also showed the best binding performance with an IF of 7.7. This proves that surface area and binding performance cannot be directly correlated. While a high surface area is a solid foundation for potential binding, the complex formation and stabilization of monomer and template cannot be overlooked in creating high-performance MIPs.

Novel approaches to studying MIPs are still emerging, and it has been demonstrated that ionic dissociation can still occur even in aprotic solvents effects. This is the predominant message in the research published by Nagy-Szakolczai et al. [61]. The manuscript outlines how conductivity experiments performed on pre-polymerization mixtures consisting of methacrylic acid and a basic template in aprotic solvents indicate that ionic dissociation is still possible. A study comparing different ratios of propranolol and methacrylic acid provides evidence of this, with higher molar ratios yielding a much greater specific conductivity. The same trend was also observed when repeating the experiment with dibenzyl amine, with the conductivity increasing with the molar ratios. Furthermore, when introducing a quaternary ammonium ion probe in toluene to the MIP, there is still evidence of ion-exchange behavior, thus bolstering their previous claims. It therefore highlights why aprotic solvents are such a common choice for MIP synthesis, with this class of porogens still facilitating the key interactions between template and monomer that enable the formation high-affinity binding sites.

Currently, we have only considered aprotic solvents for their ability to influence pore formation and polymerization efficiency. However, their environmental impact raises concerns that need to be addressed to balance their use with sustainability efforts [62]. Many of these solvents are volatile organic compounds (VOCs) that can contribute to air, soil, and water pollution if they are not properly managed or disposed of [63]. This is especially important for precipitation polymerization, as it utilizes high solvent volumes.

## 4. Protic Porogenic Solvents

Protic porogenic solvents, in contrast to their aprotic counterparts, are characterized by their ability to participate in hydrogen bonding due to the presence of hydrogen atoms bonded to electronegative atoms like oxygen or nitrogen. Examples of protic solvents include water, alcohols (such as ethanol and methanol), and carboxylic acids [64,65]. These solvents play a distinct role in the formation of porous materials, influencing both the polymerization process and the resulting pore structure. In general, protic porogens are characterized by higher dielectric constants, polarities, and *δ_H_* compared to their aprotic counterparts (Table 1). This enables strong acid–base and ionic interactions, while solvating highly polar analytes. Additionally, the hydrogen bonding capability of protic solvents can significantly affect the polymerization process with, e.g., an increase in polymerization rate [66,67,68]. When used as porogens in the synthesis of MIPs or other porous materials, these solvents’ ability to participate in hydrogen bonding can lead to the formation of materials with unique properties and functionalities, while also improving compatibility with biological systems [69]. For instance, water and alcohols can promote specific target–polymer interactions through hydrogen bonding and the stabilization of specific conformations, potentially leading to different pore structures compared to those formed with aprotic solvents [70,71]. This interaction can influence the size, distribution, and stability of the pores, which in turn affects the efficiency and selectivity of the resulting material [72]. An additional advantage is the similar swelling behavior that occurs if MIPs are prepared in aqueous media for the purpose of sensing or extraction in aqueous samples, as MIPs produced in organic solvents can run the risk of extensive or insufficient swelling in an aqueous environment, negatively affecting the rebinding performance [45]. Moreover, some protic solvents are environmentally benign and more easily recyclable compared to certain aprotic solvents, contributing to more sustainable practices in material synthesis.

However, the use of protic porogenic solvents also introduces certain challenges due to the stabilization of free radicals or the disruption of template–monomer interactions. Their ability to engage in hydrogen bonding can sometimes result in less control over the pore formation process, leading to more variable or less predictable pore structures [73]. Additionally, protic solvents such as water can be difficult to remove completely from the polymer matrix, potentially affecting the stability and performance of the final material. This can be particularly problematic in applications where precise control over the pore characteristics is crucial.

Highlighting these concepts, Zhang et al. described the creation of atrazine (ATR)-imprinted polymer layered magnetic particles through suspension polymerization with microwave heating, aimed at detecting triazines in complex samples [74]. These MIP layered magnetic particles were synthesized using methacrylic acid as the monomer, trimethylolpropane trimethacrylate and divinylbenzene as crosslinkers, AIBN as the initiator, and water as the solvent, with a small amount of toluene to dissolve all components, and with the microwave heating process lasting 120 min at 70 °C and ramping up from room temperature in 3 min. The resulting magnetic MIPs exhibited a uniform morphology and a narrow size distribution, with an average particle diameter between 100 and 200 µm, and demonstrated enhanced binding capacity and selectivity for triazines, achieving higher imprinted factors than traditional counterparts.

This methodology for analyzing triazines has both advantages and disadvantages. On the positive side, the magnetic beads facilitate an easy separation of analytes, enhancing efficiency and reducing sample loss, while their recycling through sonication and thermal revival promotes cost-effectiveness and sustainability. The approach minimizes nonspecific adsorption with n-hexane, leading to more accurate results, and the automated setup with a Shimadzu LC-2010 system (C_18_-column from Dikma, Bejing, China) and UV−Vis detector ensures precise, reproducible chromatographic measurements. However, reliance on specific bead sizes (120 to 150 μm) may limit versatility and affect separation effectiveness for different samples. Furthermore, though the synthesis utilizes environmentally friendly solvents such as water, the washing and processing steps of the MIP introduced strong VOCs which still prove troublesome. Overall, this work does emphasize how water can be used in the synthesis of high-affinity MIPs and thus breaks away from more traditional aprotic solvents.

By pursuing methods of environmentally friendlier MIP synthesis, Ostovan et al. developed a bio-based receptor towards the selective absorption of B vitamins. These MIPs were prepared using chitosan to form an imprinting network and dissolved in a 1% acetic acid solution, facilitating the formation of pre-polymerization complexes with the target vitamins shown in Figure 3 [75]. The choice of acetic acid as a protic solvent aids in the solubility of chitosan, and the subsequent elution process with methanol–water ensures the effective removal of template. The MIPs demonstrated high binding capacity, good selectivity, and fast dynamics, with optimized d-solid-phase extraction (d-SPE) conditions yielding low detection limits (1.2–5.5 μg/L) and high recoveries for vitamins B2, B3, and B6 (75.8–93.8%).

In this case, the choice of solvent selected was dictated by the polysaccharide monomer, with chitosan only being soluble in acidic conditions. As the use of other bio-based monomers (e.g., amino acids and sugars) becomes more popular, it can be assumed that water, alcohols, and even weak acids will become more common place in MIP synthesis. That said, not all bio-based monomers are water-soluble, with compounds such as fatty acids or lignin-derived monomers still requiring lipophilic solvents.

Beyond benefiting the environment, the addition of water during MIP synthesis has demonstrated improvement in the binding properties of MIPs. Foroughirad et al. highlighted this facet in their porogenic study of MIP-decorated halloysite nanotubes and their ability to bind the dye known as sunset yellow [76]. In this study, novel water-compatible molecularly imprinted polymers (MMIPs) were developed for the specific recognition of sunset yellow (SY) from aqueous solutions, using magnetically decorated halloysite nanotubes as support. The MMIPs were synthesized in water, with results showing that water improved the polymers’ adsorption properties, including higher adsorption capacity (46.4 μmol/g) and better selectivity (K_sel_ = 3.60) toward SY. The kinetic data fit well with a pseudo-second kinetic model, indicating chemisorption, while W-MMIP followed the Langmuir isotherm model and Et-MMIP adhered to the Freundlich model. The findings suggest that using water as a solvent enhances the pore structure and binding sites, making W-MMIP highly effective for recognizing SY in aqueous environments. In particular, it is believed that the presence of water improved the pore diameter and mesoporous structure present within the MIP, thus resulting in an increased binding capacity and improved recognition capabilities of the studied material.

The sol–gel process is an imprinting method that avoids free-radical polymerization and the issues this encounters in aqueous environments. This methodology offers some merits such as facilitating the room temperature construction of complexes and utilizing green solvents such as water and ethanol [77]. In detail, in sol–gel polymerization, protic solvents like water serve a double purpose of dissolving all reaction components and catalyzing hydrolysis and condensation reactions. While this can be advantageous for certain sol–gel formulations, excessive hydrogen bonding and solvation effects can lead to uncontrolled gelation and inconsistencies in the material. Therefore, careful management of the protic solvent concentration is necessary to maintain the desired gelation properties and ensure a consistent pore structure [78]. Kalogiouri et al. demonstrate this dual-purpose by applying a molecularly imprinted sol–gel towards the solid-phase extraction of bisphenol A in water, as shown in Figure 4 [77].

To this end, they synthesized a tetraethyl orthosilicate (TEOS) crosslinked APTES/PheTES-based sol–gel imprinted with BPA in an acidified isopropanol medium. The hydrolysis of both the monomer and crosslinker was catalyzed by aqueous HCl, whereas the condensation step was base-catalyzed, highlighting the critical role of solvent. The resulting MIP exhibited an excellent recovery rate (RE: 93.4 with RSD *n* = 5) and was highly reproducible and precise over the range of 90.7–96.1 ng L^−1^ of the target analyte. The accomplished limit of detection (LoD) of the setup was found to be 0.015 ng L^−1^, with a limit of quantification of 0.045 ng L^−1^. The high polarity and protic nature of the solvent system utilized in the research yielded a BPA sol–gel with an IF of 6.58, highlighting how the use of these solvents can produce polymeric systems with high specificity.

Many more examples of methodologies utilizing protic solvents for MIP synthesis exist in the literature, with these pieces of research further highlighting the impact on the field (Table 3). In summary, while protic solvents offer specific benefits, such as promoting certain interactions and potentially enhancing selectivity, they also introduce challenges that can limit the use of various polymerization methods. Balancing these effects is essential to optimize the synthesis of MIPs and achieve the desired properties and performance in the final material.

## 5. Combining Protic with Aprotic Solvents

Dual porogen systems offer a practical solution to the challenges posed by using single proticity solvents in polymerization processes. By integrating both protic and aprotic solvents, researchers can capitalize on the strengths of each type while mitigating their individual drawbacks. This approach provides enhanced control over the pore structure and morphology of the resulting MIPs, leading to improved material performance [89,90,91,92]. In detail, dual porogen systems expand precipitation polymerization possibilities and open up the possibility of emulsion polymerization, which is characterized by high control over the resulting polymer morphology [93]. The latter is composed of a continuous aqueous phase in which the organic solvent and polymerization components are emulsified with the help of a colloidal stabilizer like micelles or nanoparticles [94]. These organic droplets are stabilized by the stabilizer at the monomer–water interface, and the polymerization mainly occurs in these droplets. The size of the droplets determines the size of the formed polymer particles, enabling precise control with particle sizes tunable from hundreds of nm to tens of µm [95,96]. As previously mentioned, small particles have a higher surface-to-volume ratio, which increases the number of available binding sites [97]. Additional benefits arise when water–soluble monomers like acrylic acid or acrylamide are used because in the final polymer the functional groups are mainly found at the surface, which further increases the number of available binding sites [98]. Nevertheless, the pH of the aqueous phase needs to be controlled as the acrylate anion, which would not diffuse into the organic droplets would be predominantly present in basic solutions. In general, we will provide examples wherein dual porogen systems have benefited the performance of bulk-imprinted MIPs, precipitation polymerization MIPs, emulsion polymerization MIPs, and sol–gel MIPs [16,99,100].

Chen et al. offer an example of this with their research offering an imprinting strategy for the detection of S-naproxen (S-NAP) [101]. A free-radical monolithic imprinting approach utilizing toluene, dodecanol, EGDMA, 4-vinylpyridine, and AIBN facilitated the synthesis of an MIP directly inside a stainless-steel column. The resulting monolithic preceded to be placed within a liquid chromatography setup, where the polymer resolved a racemate of S-NAP and R-NAP. Scatchard analysis showed that two classes of binding site existed within the material, with dissociation constants estimated to be 1.045 and 5.4 μM, respectively, with the latter being towards the imprinted S-NAP. Thus, this dual porogen system yielded receptors that were highly selective towards the desired enantiomeric compound.

Porogen mixtures of different proticity can be employed in precipitation polymerization to further increase the control over the polymer morphology [102]. For example, Zhu et al. developed MIP microspheres for the high-performance liquid chromatography (HPLC)-detection of kaempferol in traditional Chinese medicine (TCM) by microwave-assisted precipitation polymerization in ACN/MeOH (4:1) [103]. Static binding experiments were conducted to assess the MIPs’ specificity towards kaempferol, with a high IF of 5.0 being achieved before employing the polymer in solid-phase extraction. Herein, a linear range of 4.5–200 mg/L, a limit-of-detection (LOD) of 8 µg per gram TCM, and kaempferol recoveries between 88.0% and 90.2% highlight the potential of the synthesized microspheres.

Liang et al. demonstrate the effects of emulsion polymerization on the morphology and enantioselectivity of a core–shell MIP towards the binding of ursodeoxycholic acid (UDCA) [104]. The porogen formulations tested through the study focused on the use of water in combination with either toluene, acetone, or dodecanol/CyOH to form an emulsion with sodium dodecyl sulfate (SDS) that can be polymerized to yield defined polymeric particles. Molecular simulations were conducted to assess the interactions between template and monomer in the different water–porogen combinations, as shown in Figure 5. It was shown that toluene as a porogen resulted in the highest complex strength of UDCA and acrylamide (AM) because toluene does not disturb hydrogen bonding.

The resulting core–shell MIPs were synthesized in the presence of water and toluene, yielding mesoporous polymers with a specific surface area of 158.83 m^2^ g^−1^, pore diameter of 7.8 nm, and a separation factor of 3.24. Furthermore, the receptor was found to have a chromatic separation of ursodeoxycholic acid and its enantiomer of 1.93, indicating that the polymer was selective towards the enantiomerically pure compound. This performance was achieved in a biphasic environment, highlighting how a two-phase emulsion polymerization in toluene and water produces water-compatible receptors. While in this research the emulsion is stabilized by the surfactant sodium dodecylsulfate SDS, the more prevalently used technique comprises Pickering emulsions, wherein the stabilization is provided by solid nanoparticles (NPs) [105,106]. This circumvents the strenuous issue of removing the surfactant from the polymer after the polymerization. For example, Wang et al. prepared MIP microspheres by Pickering emulsion for the extraction of bisphenol A from environmental samples [107]. The toluene in water emulsion was stabilized solely by Fe_3_O_4_ NPs, and the resulting MIPs showed high affinity towards bisphenol A in ethanolic solutions, with outstanding reusability over 8 cycles. Furthermore, Sun et al. developed MIPs by charged monomer polymerization in a Janus particle-stabilized Pickering emulsion for the separation of bovine hemoglobin, with the approach, particle morphologies, and rebinding performance shown in Figure 6 [96]. The particle size of the spherical MIPs was around 50 µm, with small observable cavities of ~60 nm. The resulting MIPs show an impressive IF of 3.97 and a maximal binding capacity of 385 mg g^−1^.

Beyond radical polymerizations, sol–gel polymerization benefits from dual porogen systems by improving control over the gelation process [108]. Protic solvents like water are crucial for hydrolysis and condensation reactions, while aprotic solvents can modulate the sol–gel transition and prevent issues related to excessive hydrogen bonding [16]. This combination facilitates a more controlled gelation, leading to consistent pore structures and better-defined porous networks. This concept is exemplified in the work by Moein et al. on a three-phase molecularly imprinted sol–gel-based hollow-fiber liquid-phase micro extraction combined with LC-MS [109]. In essence, a polysulfone hollow fiber was submerged in a sol solution containing 3PMTMOS (aqueous) and a template molecule in ACN. Following this, trifluoroacetic acid was used to catalyze the condensation reaction between the silane molecules, with water being added afterwards to yield the hydrolysis product. This process was found to produce materials that had excellent selectivity towards hippuric acid (a lung cancer biomarker) in plasma and urine. The system highlights how both aqueous and organic solvents enabled precise control over condensation and hydrolysis at the various steps of sol–gel formation, tailoring the sol–gel towards the LC-MS application.

The resulting modified fibers were also found to be highly robust, highly selective (against Tyr, Typ and Kyn), pH stable (between pH 3–7), and effective in various solvents (toluene, benzene, hexane, and cyclohexane, as well as aqueous environments). Beyond these few examples, many researchers are investigating porogen combinations of protic and aprotic solvents, and their applications (Table 4).

One of the primary limitations in mixed porogenic systems is the complexity involved in optimizing the porogen mixture. Different porogens may have varying solubilities, viscosities, and evaporation rates, making it challenging to achieve a consistent and reproducible polymer morphology. This variability can lead to batch-to-batch inconsistencies in imprinting efficiency and binding characteristics, impacting the reliability of MIPs in practical applications [117,118]. Achieving the desired balance between porogen composition and polymer functionality requires thorough experimental optimization, often involving multiple trial-and-error iterations. This process can be time-consuming and resource-intensive, particularly when targeting specific molecular templates with stringent recognition requirements [119].

Lastly, the use of mixed porogen systems in MIP synthesis raises several environmental implications that warrant attention, as these techniques still employ harmful solvents and additives. As the field moves towards more sustainable practices, there is a growing emphasis on developing eco-friendly porogens, such as bio-based solvents, which can minimize environmental harm and enhance the overall sustainability of MIP technologies [120]. By addressing these environmental concerns, researchers can help ensure that advancements in MIP synthesis align with broader goals of environmental stewardship and sustainability.

In summary, while mixed porogen systems offer versatility in tuning the properties of MIPs, including pore size, surface area, and binding performance, they pose challenges related to sustainability, optimization complexity, and surfactant removal.

## 6. Ionic Liquids and Deep Eutectic Solvents

The challenge of improving the performance of MIPs has spurred significant research towards new porogens for their synthesis. One class of innovative porogens are ionic liquids (ILs), which are a mixture of organic cations and inorganic anions that are liquid near room temperature [121]. To date, two extensive reviews have been published on the use of ILs in MIP synthesis, with one being focused on solid-phase extraction and the second on biorecognition and biosensing [122,123]. This review rather aims to highlight some of the benefits and applications of ILs in their use as porogens. Their unique physical and chemical properties, like high chemical and thermal stability for polymerization, strong solubility to dissolve possible targets, and a high degree of designability, have fueled their incorporation into the MIP field [124,125]. The latter results from their easily tunable composition, with prominent organic cations like pyridinium, imidazolium, or quaternary ammonium and inorganic anions like chloride, bromide, or hexafluorophosphate [126]. The changing composition in turn affects the IL’s properties drastically, as, for example, the nature of the anion affects recognition based on its hydrogen bond-accepting tendency [127]. Even though there is an extensive library of possible ILs, most of the porogens in imprinting are based on an organic solvent like chloroform in combination with an IL composed of imidazolium with either BF_4_^−^ or PF_6_^−^ [128,129,130]. In comparison to protic solvents, the commonly used IL1-butyl-3-methylimidazolium hexafluorophosphate ([BMIM]PF_6_) has a lower dielectric constant but similar polarity, which favors the ion pairing of strongly polar molecules (Table 1). Even though the mechanisms behind the improvement of the binding performance are still yet unclear, some common advantages that were observed are better performance in aqueous environments, minimized polymer swelling, reactant stabilization, increased polymerization speed, and most notably increased affinity to the target [131,132,133,134]. A variety of MIPs synthesized in ILs can be seen in Table 5. He et al. were the first to use an IL without an organic solvent to develop a testosterone sensor based on imprinted silica. In the first step, a testosterone-monomer complex was synthesized, which was later cross-linked by tetraethyl orthosilicate in [BMIM]BF_4_ and aqueous HCl [135]. It was shown that the sensor had high affinity towards testosterone compared to a reference material with an IF around 10. Furthermore, testosterone propionate was employed to investigate the sensor’s selectivity, and low cross-reactivity was found. Nevertheless, the sensor was not tested in a real-life application, limiting its possible impact. More commonly, binary or ternary porogen mixtures were used, for example, by Deng et al., who designed a rutin-imprinted polymer with DMF-ACN-[BMIM]PF_6_ as the porogen [130]. This is commonly applied to lower the viscosity of ILs and adjust polar and dielectric properties. The imprinted material was compared to the same material with the omission of the IL during the synthesis. The superiority of the IL-based material was evident as a higher specific surface area, higher adsorption capacity, higher selectivity, and an increase in IF from 2.74 to 4.84, as seen in Figure 7. The material was employed in the solid-phase extraction of rutin from Tartary buckwheat, and the purity of the extract was above 92% in a single cycle.

Similarly, Sun et al. developed MIP monoliths with a porogen composed of DMSO and [BMIM]BF_4_ for the extraction of chlorogenic acid from *Eucommia ulmoides* leaves [136]. Even though no reference MIPs without the IL were synthetized, the high column permeability (1.53 × 10^−13^ m^2^), an IF around 10, and the short separation time and efficiency prove the excellent performance of the IL-based polymer. Furthermore, a high recovery of 89% could be achieved from the leaves with direct loading onto the column without pretreatment.

The same IL was utilized in the synthesis of MIPs for chicoric acid by Sun et al. [137]. In this case, [BMIM]BF_4_ was combined with DMSO and DMF, while the target affinity was improved by the addition of Zn(II) as a metallic pivot. After the tedious optimization of various synthesis parameters, the best performing polymer had an impressive IF of 24.81. In the solid-phase extraction of chicoric acid from *Cichorium intybus* L. leaves, the MIPs yielded a recovery of 77.5%, with an excellent CA purity of the extracts of 98%.

The table highlights a major drawback of ILs in that a limited number have been investigated so far. Only [BMIM]PF_6_ and [BMIM]BF_4_ have been regularly used in imprinting, while the library of aprotic solvents is significantly larger. Additionally, co-solvents are regularly employed to tackle ILs’ high viscosity, which results in a highly complex porogen mixture. The addition of potentially carcinogenic solvents like DMF or toluene reduces their eco-friendliness significantly [143,145,146,147]. Lastly, as concerns about the biodegradability and toxicity of ILs have risen, researchers have turned their attention to greener alternatives like deep eutectic solvents (DESs) [148]. Besides better biocompatibility, DESs do not need costly purification steps, and starting materials are generally more affordable, with a larger library of investigated DES in imprinting is available [149,150]. In terms of solvent properties, DES are more polar than ILs and have a higher tendency for hydrogen bonding (Table 1). A comprehensive review on DESs in molecular imprinting was published by Madikizela et al., highlighting their versatility as template, functional monomer, cross-linker, porogen, and modifiers [151]. DESs are a combination of a hydrogen bond donor (HBD) with a hydrogen bond acceptor (HBA), which lowers the melting point of the eutectic mixture compared to the individual components. From the four types of DESs, type III is the one predominantly used in molecular imprinting, with choline chloride (ChCl) being the HBA [151,152]. Precise porogen acidity control for imprinting can be obtained by varying the nature and molar ratio of HBD, with alcohols increasing pKa and acids lowering it [153,154]. For the DES synthesis, multiple straight-forward techniques have been employed based on heating and stirring, microwave irradiation, ultrasound, or rotary evaporation [151,155,156]. Nevertheless, similarly to ILs, DESs are mostly employed as a ternary mixture with other organic porogens, as a high amount of DESs could result in high viscosity and steric hinderance [157,158]. Interestingly, compared to traditional imprinting, DES-based MIP synthesis employs more protic solvents like MeOH, EtOH, or water, even though traditionally they are thought to hinder imprinting due to the hydrogen bond donating or accepting abilities [159,160,161]. The resulting materials share common advantages over their traditional counterparts like improved hydrophilicity, surface area, adsorption capacity, selectivity, and IF, and an overview of DES-based MIPs can be found in Table 6 [151]. As an example, Meng et al. developed MIPs for the microextraction of the antibiotic levofloxacin, with the most commonly used DES ChCl–ethylene glycol (EG) as the porogen [162]. The performance of DES-MIP was compared to MIP with MeOH as the porogen, with the former demonstrating a larger surface area (111 vs. 6.9 m^2^/g), pore volume, and diameter and higher maximal binding capacity (*Q*_max_(DES) = 2.16 mmol/g vs. *Q*_max_(MeOH) = 1.11 mmol/g) and affinity (IF(DES) = 1.8 vs. IF(MeOH) = 1.2) towards levofloxacin. The DES-based polymer was able to extract levofloxacin from plasma samples from hospitalized patients, with recoveries of 95.3 to 99.7%. Hydrophilic-imprinted resin was synthesized by Liang et al. in a ternary mixture of water and a DES-based on ChCl and 1,4-butandiol (1,4-BD) (1:2) to extract kaempferol from tea [163]. The hydrophilicity of the DES-based material was characterized by contact angle measurements, showing a low contact angle of 34.8°. Furthermore, the extraction performance was compared to MIPs synthesized in ACN and tetrahydrofuran (THF), with a slightly higher LOD but improved recovery percentages. Another hydrophilic-imprinted resin (HIR) based on the same DES–water as the porogen was designed by Wang et al. for the recognition of two tumor biomarkers, namely vanillylmandelic acid (VMA) and homovanillic acid (HMA) [164]. The hydrophilic resin exhibited excellent IFs of 111 towards VMA and 19 towards HMA, with maximal binding capacities around 20 mg/g for both biomarkers. Furthermore, combination with a HPLC-UV detector resulted in a method LOD and LOQ of 0.5–1.0 ng/mL and 1.5–3.5 ng/mL, respectively. Compared with previous methods, a lower LOD, wider linear range, and higher precision was obtained. The synthesis approach, selectivity investigation, and comparison with other commercially available adsorbents can be seen in Figure 8.

Besides polymerizations in solutions, Porfivera et al. utilized a natural DES based on glucose, citric acid, and water (1:1:6) as the porogen to electropolymerize thionine onto a screen-printed carbon electrode to detect epinephrine in saliva samples [165]. The readout was based on the variation in change transfer resistance in the presence of a redox probe determined by electrochemical impedance spectroscopy. Even though other sensors based on multiple-step synthesis had lower LoDs, the DES-based sensor had a linear range of more than three orders of magnitude in the clinically relevant range, with a low sensor repeatability of 6.0% for six individual sensors. Furthermore, the real-life applicability was thoroughly tested, as human saliva, two pharmaceutical epinephrine preparations, a glycerol aqueous solution, and a mixture of epinephrine, norepinephrine, and serotonin were investigated with satisfactory recoveries ranging from 80.0 to 102.8%. Lastly, a ternary mixture of IL, DES, and DMF was employed to extract cetirizine from ethanolic solutions by Wei et al. [166]. In detail, DMF was mixed with [BMIM]BF_4_ and ChCl:EG (1:6.8 (wt%)) as a porogen and Co^2+^ was used as a metallic pivot. Even though it seems extensive to utilize five different compounds as a porogen, the resulting polymer had very high affinity towards the target, with a maximal IF of 31.84. After optimization of the SPE parameters, cetirizine was selectively extracted from an ethanolic solution with a recovery of 97.8%, but no complex samples were analyzed. Besides all the merits, a challenge that remains is the development of new DESs, since the HBA:HBD donor needs to be empirically tested to determine which composition results in a liquid. Furthermore, the eutectic mixture might have a melting point lower than its components, but it is often above room temperature, which limits its ease-of handling and potential applications [157].

**Table 6 polymers-17-01057-t006:** Summary of MIPs based on DES with co-solvents as porogens and their IFs and applications.

Template	DES (HBA:HBD)	Co-Solvent	Reaction Type	Application	IF	Ref.
Levofloxacin Tetraclyne	Betaine–EG (1:2)	water	Condensation and Polymerization	Sensing	1.4–2.2 *	[161]
Thionine	Glucose–Citric Acid (1:1)	water	Electro-polymerization	Sensing	n.d.	[165]
Triazines	L-menthol–Formic acid (1:1)	/	Bulk polymerization	Extraction	2.5 *	[167]
Kaempferol	ChCl: 1,4-BD (1:2)	water	Condensation	Extraction	2.7	[163]
Vanillylmandelic acid Homovanillic acid	ChCl: 1,4-BD (1:2)	water	Condensation	Extraction	19.2–111.6	[164]
4,4′-dichlorobenzhydrol	ChCl: 1,4-BD (1:2)	water	Condensation	Extraction	9.1	[168]
Naproxene	ChCl–1-butylimidazole (1:1)	ACN	Bulk polymerization	Extraction	1.7	[169]
Organophosphates	ChCl: Glycerol	70% EtOH	Polymerization	Extraction	4.5	[170]
Clorprenaline Bambuterol	ChCl:EG (1:2)	/	Condensation	Extraction	~6 *	[171]
Lysozyme BSA	ChCl:EG (1:2)	water	Polymerization	Extraction	n.d.	[172]
Quinolones	ChCl:EG (1:2) ChCl: TMAC (1:2) ChCl: TMAB(1:2)	water	Condensation	Extraction	n.d.	[173]
Fenbufen	ChCl:EG (1:2)	[BMIM]BF_4_	Bulk polymerization	Extraction	3.9	[133]
Levofloxacin	ChCl:EG (1:2)	/	Bulk polymerization	Extraction	1.8	[162]
Cetirizine	ChCl:EG (1:6.8)	[BMIM]BF_4_ and DMF	Metallic pivot radical polymerization	Extraction	31.5	[166]

TMAB = tetramethylammonium bromide, TMAC = tetramethylammonium chloride, n.d. = not determined, IFs marked with * were estimated from the rebinding data.

## 7. Emerging Porogenic Systems

Another approach is to circumvent liquid porogens completely and instead employ solid porogens in mechanochemistry. Up to now, three publications on the synthesis of MIPs have been published, with Furtado et al. being the first [174,175,176]. In the latter, atenolol-imprinted polymers were synthesized by liquid-assisted grinding (LAG) and compared to solvent-based MIPs. In LAG, small amounts of solvent (0–2 µL/mg) are added to minimize termination reactions and accelerate the reaction rate [177]. In this case, for the polymerization, ~0.1 µL/mg of ACN and catalytic amounts of alumina were added, and the polymerization was irradiated by UV light to ensure initiation, with the process shown in Figure 9.

The resulting mechanosynthesized MIPs showed high affinity (IF = 9.84) towards atenolol in comparison to the classical solvent-based MIPs (IF = 1.2). Furthermore, the best performing MIP showed high selectivity towards atenolol in comparison to other β-blocker analogs, but the performance was not investigated in a real-life sample. The first MIPs developed by mechanochemistry with salt-assisted grinding were employed for the extraction of L-leucine [174]. In particular, NaCl was added at two times the weight of the remaining components, and the polymerization was carried out for 6 h. The resulting MIPs had average diameters of 1–3 µm, and physisorption revealed a meso- and macroporous structure. The MIPs revealed a maximal IF of 1.98 in water, showing good affinity towards L-leucine, but the real-life applicability was not assessed. In the same publication, another green imprinting strategy was applied, namely imprinting in supercritical CO_2_ (scCO_2_). In terms of greenness, scCO_2_ is non-toxic, non-flammable, inert, and can easily be removed, making it a promising alternative [178]. Its physical properties are defined by a very low dielectric constant and polarity, which is favorable for not disrupting template–monomer complexes [34]. Major advantages of using scCO_2_ in imprinting are its aprotic nature and the ease of extraction of the template and unreacted reactants after the synthesis. While traditional MIPs often require extraction by organic solvents for long periods of time, in scCO_2_-assisted synthesis, flushing the reactor with fresh CO_2_ for a few hours suffices [179]. Further reducing the synthesis time is the fact that after extraction, the polymers are ready to use without any further drying or grinding. For a thorough overview on scCO_2_ in the molecular imprinting of molecules, we recommend the review by Furtado et al. [180]. As previously mentioned, MIP synthesis assisted by either mechanochemistry or scCO_2_ was investigated for its affinity towards L-leucine [174]. In comparison with the mechanochemistry-assisted MIPs, the scCO_2_-MIPs showed an improved surface area (90 vs. 6.8 m^2^/g) and smaller mesopore diameters (21 vs. 34 nm) at similar particle sizes, which in turn resulted in higher binding capacities and a greatly enhanced IF of 12. The first reported use of scCO_2_ in imprinting was as early as 2006 for the controlled drug delivery of salicylic and acetylsalicylic acid [181]. The employed porogen was a mixture of scCO_2_ and a carboxylic acid end-capped perfluoropolyether oil as stabilizer, with the imprinted materials being able to adsorb significantly more drugs than their non-imprinted counterparts. Similarly, more drugs could be released from the imprinted materials over a longer period of time, but further adsorption characterizations of the material are limited due to the limited application in drug release. More recently, Viveiros et al. developed an extraction procedure of acetamide, a pharmaceutical impurity, by scCO_2_-assisted MIP synthesis and template desorption [182]. In this work, MIPs synthesized with pure CO_2_ as the porogen were compared to MIPs with the addition of small amounts of ACN as a co-solvent and a previous study based on pure organic porogen [183]. Both scCO_2_-assisted pathways resulted in uniform MIP particles, with an average diameter of ~5 µm and surface areas between 16 and 34 m^2^/g, highlighting the good control of MIP morphology of scCO_2_ as a porogen. From the three compared porogens, pure scCO_2_ resulted in the best performing methacrylamide–MIP with, a maximal IF of 4.45 and an impressive binding capacity of 2.21 mmol/g. The selectivity was investigated but unfortunately no extraction from a pharmaceutical product was assessed. An example of a scCO_2_ synthesis setup with a morphology comparison between bulk and scCO_2_ MIPs can be seen in Figure 10.

One of the drawbacks of scCO_2_ is the need for specialized equipment like pumps and autoclaves, which are not inexpensive compared to traditional MIP synthesis, which often relies solely on vials and a heating source [53,185].

Another novel porogen is ice, as imprinted cryogels have recently been developed for multiple templates ranging from small molecules to proteins and enzymes [186]. In general, the polymerization components are dissolved in water with a surfactant and the reaction is initiated in sub-zero temperatures by an oxidation–reduction initiator [187]. The ability to form pores stems from the fact that most of the available space is occupied by ice, while polymerization only occurs in the micro-liquid phase within the ice. Upon thawing of the ice, this porous structure stays intact, producing polymers with macro-, meso-, and micropores, whose distribution can be tuned by varying the ratio of monomer and cross-linker. Further advantages are low-pressure drop, very short adsorption times, short diffusion times, high water compatibility, and reusability [188]. For example, Kartal et al. developed a cholesterol-imprinted cryogel for the removal of cholesterol from milk samples [189]. The hydrophilicity of the cryogels was comparable to hydrophilic poly(HEMA) particles, and the particles had a high specific surface area of 17.6 m^2^/g. The cryogels are predominantly macroporous, which is in contrast to previously mentioned polymers that are mostly mesoporous. In adsorption experiments, a very high maximal binding capacity of 288.7 mg/g was determined and a high selectivity towards cholesterol was demonstrated. Additionally, cholesterol could be removed from spiked milk samples with recoveries of 94.3–100.3% at cholesterol concentrations up to 0.5 mg/mL. Furthermore, Bakhshpour et al. developed tyrosine-imprinted cryogels based on two different monomers, which performed better than MIPs prepared by suspension polymerization and an imprinted membrane [188,190]. The best performing MIP was based on HEMA and a N-methacryloyl-L-histidine-Cu(II)-tyrosine complex as the template, and characterization showed a high specific surface area of 39 m^2^/g and a high swelling ratio of 82.4%, owing to its hydrophilicity. After optimization of the adsorption conditions, a maximal binding capacity of 84.44 mg/g and an IF of ~6 were achieved, which outperforms previously published tyrosine-imprinted materials. Owing to the previously mentioned water compatibility and pore size designability, a lot of imprinted cryogel research in the literature focuses on macromolecules like immunoglobulin G (igG) [191]. Herein, the same monomer as in the previous publication was combined with N-isopropylacrylamide (NIPAM) to yield a thermoresponsive cryogel for the purification of igG from human plasma, which can be seen in Figure 11. As the resulting material is thermoresponsive, the influence on igG binding at different temperatures was investigated, and a higher affinity was observed above the lower critical solution temperature (LCST) of NIPAM. This is explained by the expulsion of water above the LCST, increasing the hydrophobic nature of the polymer and favoring the binding of igG. Most notably, the cryogel was employed to extract igG from human plasma samples, with a recovery of 24%, and the reusability was assessed for 10 cycles, with a negligible loss of binding capacity after each cycle.

Another showcase of the advantages of imprinting in ice is an electrochemical sensor for insulin developed by Wardani et al. [192]. In this work, carboxylated multiwalled carbon nanotubes were drop-casted onto a gold electrode, which was further functionalized by the cryo-polymerization of chitosan and acrylamide directly on the electrode’s surface. The insulin detection was investigated using square-wave voltammetry, with high insulin selectivity, low cross-reactivity, a very low LOD of 33 fM, and a linear range of 0.05–1.4 pM. Furthermore, the sensor was investigated for its ability to detect insulin in blood serum samples by comparison to a commercially available electrochemiluminescence immunoassay. The sensor displayed satisfactory recoveries of 90–111% and was astonishingly stable for 46 uses over 10 weeks, outperforming previously published insulin sensors. One last class of porogens that are employed to improve the hydrophilicity of MIPs are metal–organic gels (MOGs), also called coordination polymers, with three publications to date [193]. MOGs are composed of metal ions and organic ligands, which form a sort of polymer through coordination bonding, hydrogen bonding, or other intermolecular forces [194]. Their advantage in MIP synthesis results from their inherent porogenity, tunable design by selection of the metals and ligands, and excellent solubility. The first MOG-based MIPs were developed by Ma et al. for drug delivery of the antibiotic levofloxacin [193]. The polymerization was carried out in a mixture of ethanol and a MOG composed of Fe^3+^ and 1,3,5-benzenetricarboxylic acid (H_3_BTC), with the control MIPs omitting the MOG. The polymer structure was a porous “imprint” of the MOG, with the resulting mesoporous MIPs having a more than 100 times higher surface area (288 vs. 2.7 m^2^/g), larger total pore volume, and smaller pore diameter. This, in turn, resulted in a higher maximal binding capacity and IF in comparison to pure-ethanol MIPs. The MIPs were applied for the drug release of levofloxacin, and while the NIP released 85% of the drug in the first hour, the MIP could steadily release the drug for up to 12 h. Another proof of the improved aqueous behavior of MOG-based MIPs was shown by Wan et al., who imprinted sildenafil and compared the material’s performance in different rebinding solvents [195]. The MIPs showed the highest affinity towards the target in aqueous solution compared to common extraction solvents like methanol, can, and ethanol. Lastly, Zhao et al. developed a single-walled carbon nanotube (MWCNT)-based MIP composite for the drug release of the anti-tumor drug aminoglutethimide (AG), based on ethanol and Fe-H_3_BTC as the porogen [196]. The optimized composite material had an IF of 2.5 compared to an IF of 1.48 for MWCNT-free MIPs. Most importantly, the material did not show any cytotoxicity to MCF-7 cells, and in vivo drug release studies showed better performance than when administrating the drug alone. While the highest plasma concentrations of pure AG were reached after 2 h and rapidly decreased afterwards, the MWCNT-MIPs resulted in high AG plasma concentrations up to 5 h.

## 8. Conclusions

In conclusion, the selection of an appropriate porogen is a critical determinant of the performance of MIPs, influencing their morphology, surface area, swelling behavior, and binding efficiency. For developing novel MIPs, aprotic and protic porogens can serve as a good starting point for either organic or aqueous applications, respectively. The incorporation of non-traditional porogens can enhance surface area, pore size, hydrophilicity, etc., which, in turn, improves binding performance. In detail, traditional aprotic solvents remain widely employed due to the large available literature, their ability to produce well-defined porous structures, and their non-interference in hydrogen bonding. Due to their versatility in polarity and ion stabilization, the right choice of porogen can strongly complement the desired template–monomer interaction. However, their environmental impact and limited compatibility with aqueous applications restrict their broader utility. In contrast, protic solvents enhance the hydrophilicity and aqueous performance of MIPs, improving performance in biological and environmental applications. Additionally, protics enable sol–gel imprinting, further enhancing the versatility of imprinting techniques. The challenge with these solvents lies in their potential to disrupt hydrogen bonding interactions, which can compromise template recognition and lead to less efficient imprinting. Combining aprotic and protic solvents in emulsion polymerization allows for precise control over particle size and a high degree of chemical functionalities at the polymer surface. This, in turn, improves the specificity and selectivity of the developed MIPs. Nevertheless, high amounts of solvent are necessary, and surfactants can be hard to remove from the polymer matrix. In search for more environmentally friendly porogens, imprinting in ILs and DES has become a common approach due to their tunable properties, with resulting MIPs having improved hydrophilicity, surface area, and target affinity. Their high viscosity, empirical development, and cost of co-solvents limits their widespread adoption. Furthermore, novel porogens that differ from traditionally understood solvents, namely solids in mechanochemistry, ice in cryogels, MOGs, and scCO_2_-based imprinting, have resulted in high-affinity materials for numerous applications, which outperform their solvent-based counterparts.

## Figures and Tables

**Figure 1 polymers-17-01057-f001:**
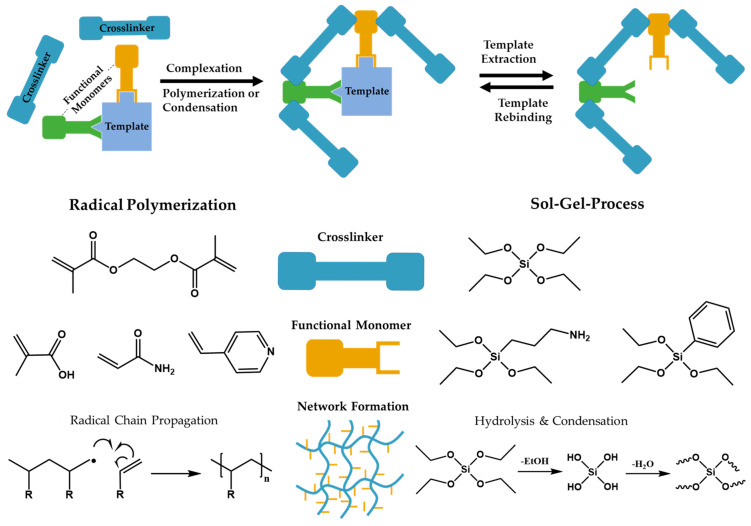
(**Top**): Schematic of MIP synthesis and rebinding process and (**bottom**): crosslinkers (blue), functional monomers (orange), and network formation for radical polymerization and sol–gel synthesis of MIPs (wavy lines represent silica network).

**Figure 2 polymers-17-01057-f002:**
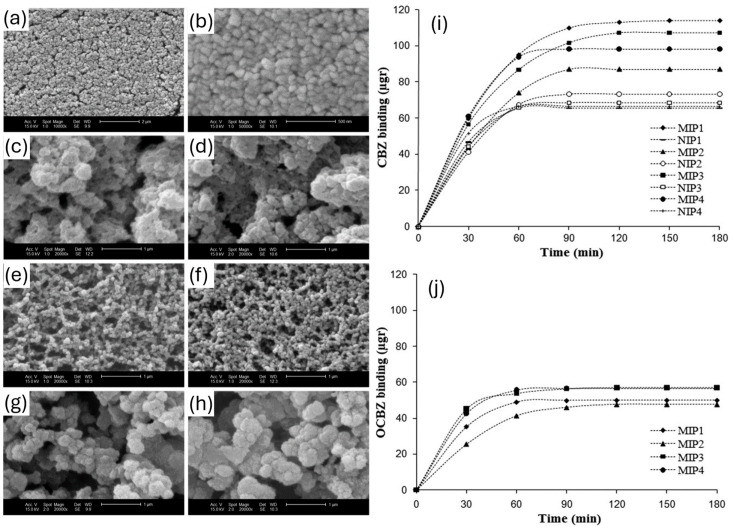
(**left**) SEM images of MIP/NIP nanopsheres synthesized in different porogens ACN (MIP (**a**), NIP (**b**)), CHCl_3_ (MIP (**c**), NIP (**d**)), ACN:CHCl_3_ (MIP (**e**), NIP (**f**)), and toluene (MIP (**g**), NIP (**h**)); (**right**) adsorption behavior of MIPs and NIPs towards the target CBZ (**i**) and an analog oxcarbazine (OCBZ) (**j**). Reprinted from [18] with copyright from Elsevier.

**Figure 3 polymers-17-01057-f003:**
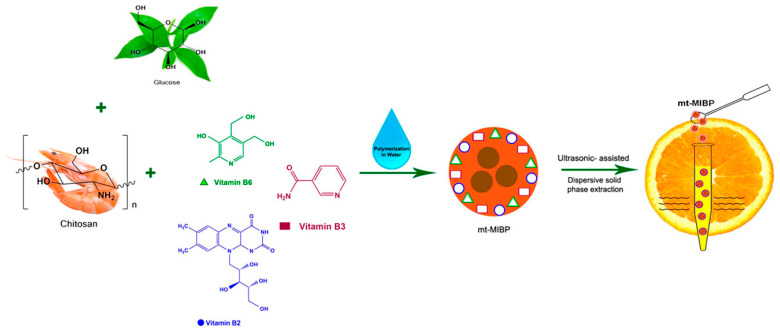
The synthesis of chitosan-based MIPs towards the detection of B-vitamins utilizing polymerization techniques in water. Reprinted from [75] with copyright from ACS publications.

**Figure 4 polymers-17-01057-f004:**
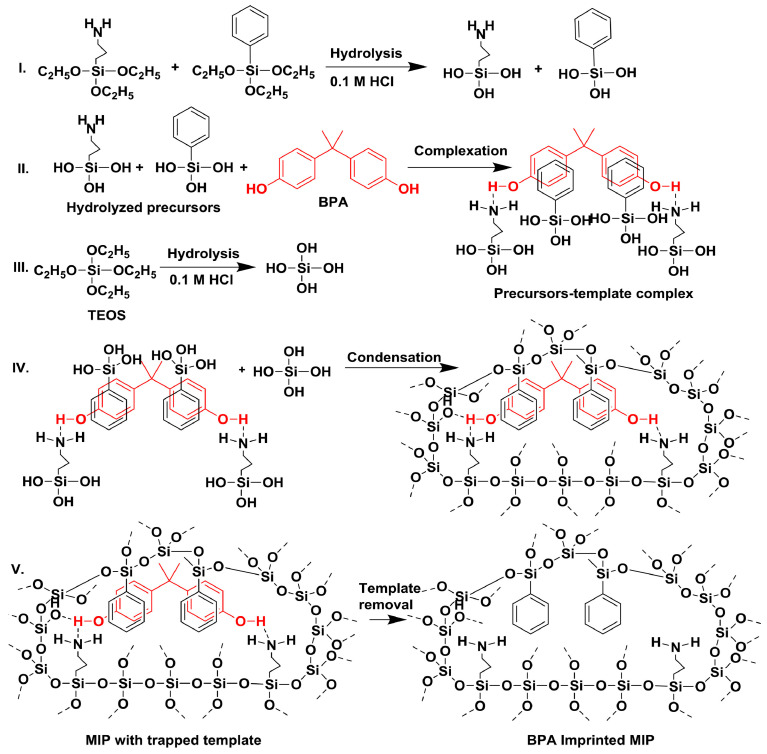
Different steps involved in BPA (red molecule)-templated molecularly imprinted polymer synthesis. (**I**) Hydrolysis of functional monomers, (**II**) complexation of monomers and BPA, (**III**) hydrolysis of TEOS, (**IV**) condensation of monomers and TEOS around BPA and (**V**) removal of BPA from MIP. Reprinted from [77] with copyright from Elsevier.

**Figure 5 polymers-17-01057-f005:**
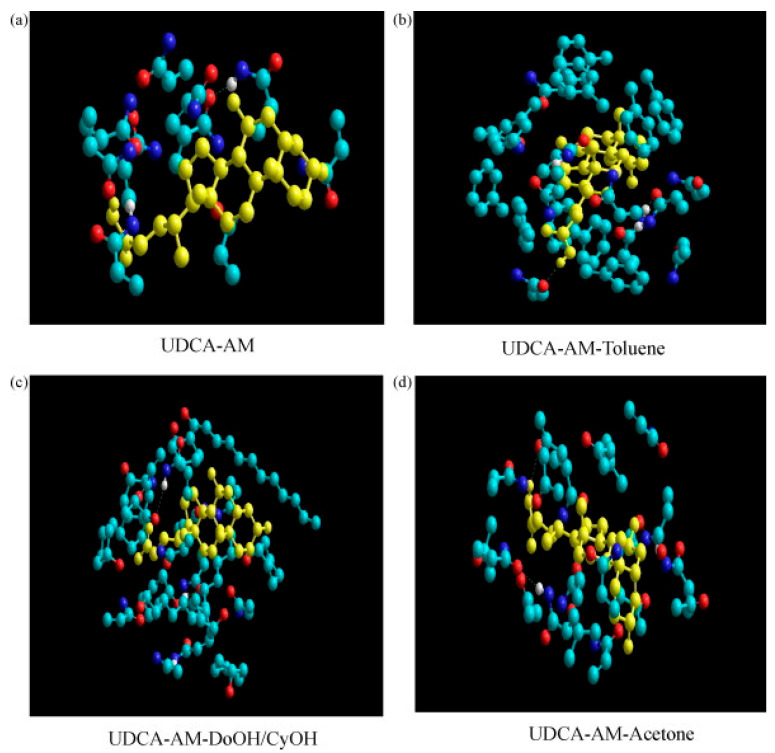
Molecular simulations of template (yellow)–monomer complexes in different porogens; ((**a**) no porogen, (**b**) toluene, (**c**) DoOH/CyOH, (**d**) acetone). For monomer and porogen, C-atoms are light blue, N-atoms are dark blue and O-atoms are red. Reprinted from [104] with copyright from Elsevier.

**Figure 6 polymers-17-01057-f006:**
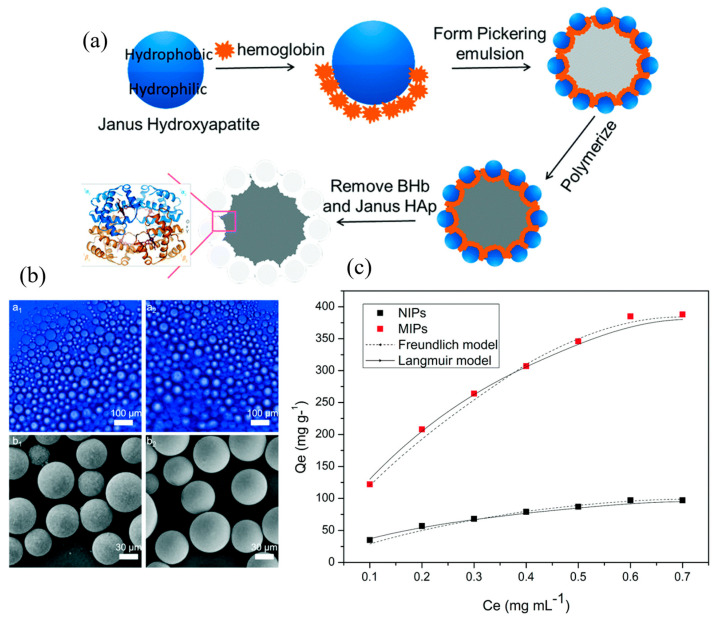
(**a**) Approach to MIP synthesis using Janus particle-stabilized Pickering emulsion, (**b**) freshly prepared Pickering emulsion (**a1**) and after standing at room temperature for 8 h (**a2**), MIP particle morphology (**b1**) and NIP particle morphology (**b2**). (**c**) Bovine hemoglobin rebinding performance of MIPs fitted by Freundlich and Langmuir model. Reprinted from [96] with copyright from RSC Publishing.

**Figure 7 polymers-17-01057-f007:**
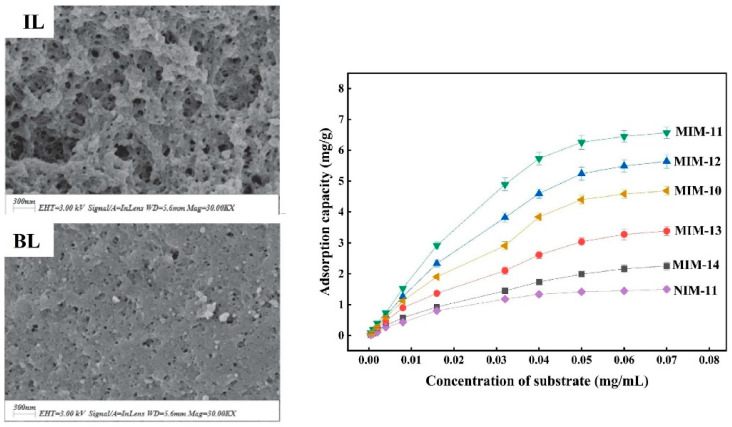
(**Left**) SEM images of IL-based MIPS and bulk (BL)-based MIPs; (**right**) adsorption study of IL-based MIPS (MIM/NIM-11) and bulk-based MIPs (MIM-10). Reprinted from [130] with copyright from Elsevier.

**Figure 8 polymers-17-01057-f008:**
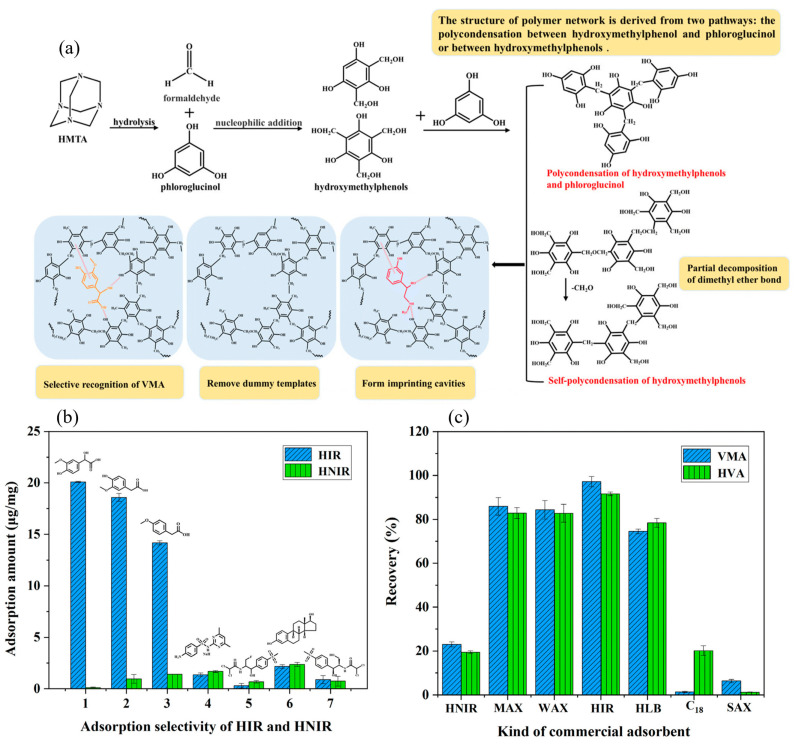
(**a**) Imprinting approach using condensation of formaldehyde and phloroglucinol, (**b**) selectivity of HIR (blue) and HNIR (green) towards VMA (1), HMA (2), and other similar compounds, and (**c**) comparison between developed HIR with commercial adsorbents for recovery of VMA (blue) and HVA (green). Reprinted from [164] with permission from RSC Publishing.

**Figure 9 polymers-17-01057-f009:**
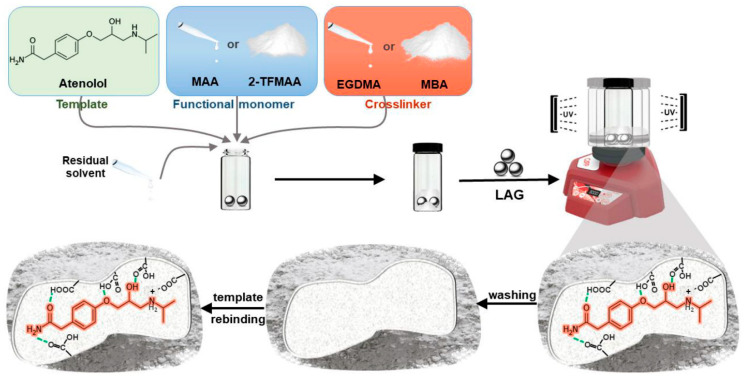
Mechanosynthesis of atenolol-imprinted polymers. Reprinted from [175] with copyright from MDPI.

**Figure 10 polymers-17-01057-f010:**
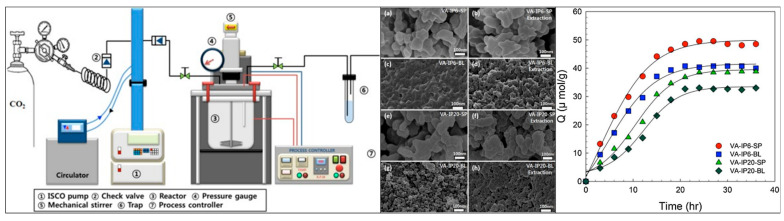
(**left**) Supercritical fluid-assisted polymerization apparatus for the preparation of MIPs, (**middle**) SEM images and of VA-IPs with 6.0 mol and 20 mol EGDMA prepared using supercritical fluid assisted polymerization (SP) and bulk polymerization (BL) before/after removal templates. (**a**) VA-IP6 prepared using SP, (**b**) VA-IP6 removed VA as the template (SP), (**c**) VA-IP6 prepared using BL, (**d**) VA-IP6 removed VA as the template (BL), (**e**) VA-IP20 prepared using SP, (**f**) VA-IP20 removed VA as the template (SP), (**g**) VA-IP20 prepared using BL, (**h**) VA-IP20 removed VA as the template (BL) and (**right**) adsorption behavior of Bl- and SP-imprinted polymers. Reprinted from [184] with copyright from Elsevier.

**Figure 11 polymers-17-01057-f011:**
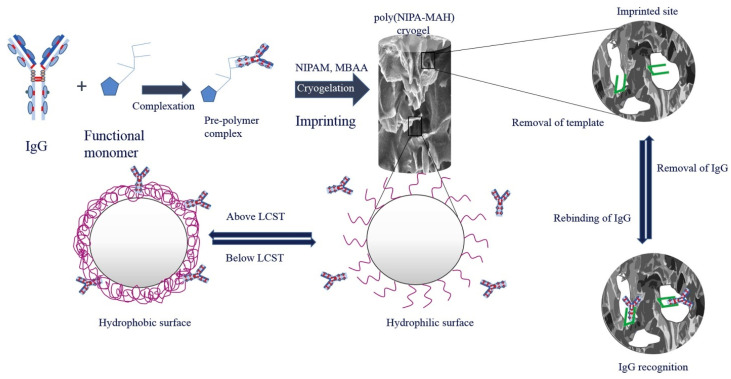
Schematic representation of cryogelation and recognition mechanism of thermoresponsive igG-imprinted polymers. Reprinted from [191] with permission from Elsevier.

**Table 1 polymers-17-01057-t001:** Boiling points, dielectric constants, Hansen solubility parameters, and polarities of different solvents (■ aprotic solvents, ■ protic solvents, ■ emerging solvents) used in MIP synthesis.

Solvent	*bp* [°C]	Dielectric Constant	Hansen Solubility Parameter [MPa^1/2^]			Polarity *E_T_* (30) [kcal/mol]
			*δ_d_*	*δ_p_*	*δ_H_*	
Acetone	56.3	21	15.5	10.4	7.0	42.2
ACN	81.6	37.5	15.3	18	6.1	45.6
Chloroform	61.2	4.8	17.8	3.1	5.7	39.1
DCM	39.8	8.9	18.2	6.3	6.1	40.7
DMF	153	38.3	16.8	11.5	10.2	43.2
DMSO	189	46.4	18.4	16.4	10.2	45.1
THF	66	7.6	16.8	5.7	8	37.4
Toluene	111	2.4	18	1.4	2	33.9
EtOH	78.3	34.6	15.8	8.8	19.4	51.9
MeOH	64.7	33.6	15.1	12.3	22.3	55.4
Water	100	80.4	15.5	16	42.4	63.1
[BMIM][PF_6_]	180 *	16.1	17.9	9.8	8.1	49.0–53.2
EG:ChCl (1:2)	439	-	15.9	5.3	19.5	57.3
scCO_2_	-	1.03–1.6	15.6	5.2	5.8	34.5

* decomposes.

**Table 2 polymers-17-01057-t002:** Selection of MIPs based on aprotic solvents with different polymerization approaches and applications.

Target	Porogen	Polymerization Approach	Application	IF	Ref.
Nicotinamide	ACN Chloroform Toluene	Monolithic bulk free-radical	Solid-phase extraction	n.d.	[44]
Ibuprofen	DMF	Monolithic bulk free-radical	Solid-phase extraction	~8 *	[45]
2-phenylproponic acids	DMF	Precipitation	Solid-phase extraction	1.3–3.3	[46]
Progesterone, testosterone	Toluene Chloroform ACN	Monolithic bulk free-radical	Solid-phase extraction	2.4–3.1	[47]
Terbutylazine	Toluene	Monolithic bulk free-radical	Solid-phase extraction	n.d.	[48]
Pinacolyl methylphosphonate	Toluene and ACN	Precipitation	Sensing	~1.3 *	[49]
Tetracycline	Chloroform and ACN	Precipitation	Drug release	5.1	[50]
Diisopropylurea	Dichloromethane	Monolithic bulk free-radical	Solid-phase extraction	~2 *	[51]
Estradiol	Toluene	Monolithic bulk free-radical	Solid-phase extraction	4.8	[52]
Xylazine	Toluene and chloroform (3:1)	Monolithic bulk free-radical	Sensing	2.0	[53]
Pyocyanin	Chloroform	Monolithic bulk free-radical	Sensing	1.6	[17]
Amphetamine	DMSO	Monolithic bulk free-radical	Sensing	4.4	[54]
(S)-N-Butyryl homoserine lactone	DMSO	Monolithic bulk free-radical	Sensing	2.8	[55]
Folic acid	DMSO–ACN (5:3)	Monolithic bulk free-radical	Solid-phase extraction	4.0	[56]
Quercetin	Acetone	Monolithic bulk free-radical	Solid-phase extraction	8.2	[57]

n.d. = not determined, IFs marked with * were estimated from the rebinding data.

**Table 3 polymers-17-01057-t003:** Selection of MIPs based on polymerizations and condensation reactions in protic solvents and their IF and applications.

Template	Porogen	Approach	Application	Additive	IF	Reference
Triazines	Water	Radical polymerization	Solid-phase extraction	5% toluene	2.7–4.7	[74]
Sunset Yellow	Water	Radical polymerization	Solid-phase extraction	-	1.33	[76]
Atenolol	Butanol or propanol	Bulk or precipitation polymerization	Solid-phase extraction	-	4.2 and 11.7	[79]
Histamine	EtOH	Precipitation polymerization	Sensing	-	2.3	[80]
Synephrine	Methanol/water (4:1)	Precipitation polymerization	Solid-phase extraction	-	~2 *	[64]
Triazines	EtOH/water (9:1)	Radical polymerization on silica particle	Solid-phase extraction	Poly-vinylpyrrolidone	n.d.	[81]
Gallic acid	Water	basic polymerization	Solid-phase extraction	Phosphate buffer	1.7	[82]
B-vitamins	Water/acetic acid (99/1)	Condensation	Solid-phase extraction	1 M NaOH	~3–4 *	[75]
Bisphenol A	Acidified isopropanol	Sol–gel approach	Solid-phase extraction	1 M NH_4_OH	6.6	[77]
2,4-Dichlorophenoxy-acetic acid	EtOH/water (10:3)	Sol–gel approach	Solid-phase extraction	Conc. HCl	1.5	[83]
Hydrochlorothiazide	Water	Sol–gel approach	Analyte monitoring	CTAB and NH_4_OH	~4.5 *	[71]
1-naphthyl phosphate	Water/EtOH (5:3)	Sol–gel approach	Solid-phase extraction	-	32.2	[84]
Folic Acid	Water	Sol–gel approach	Sensing	NH_4_OH	2.2	[85]
Bisphenol F	Water	Sol–gel approach on electrode	Sensing	CTAB and NH_3_	~6 *	[86]
Creatinine	Water	Sol–gel Approach	Sensing	1 M HCl and Al_3_Cl_3_	2.4	[87]
Salicylic acid	EtOH/water (4:1)	Sol–gel approach	Drug release	0.1 M HCl	9.0	[88]

n.d. = not determined, IFs marked with * were estimated from the rebinding data.

**Table 4 polymers-17-01057-t004:** Selection of MIPs based on combining aprotic and protic porogens and their performance indicators and applications.

Template	Porogen	Approach	Additive	Application	IF	Particle Size	Ref.
(S)-Naproxen	n-dodecanol/toluene	Bulk	-	Enantiomeric Separation	n.d.	n.d.	[101]
Ursodeoxycholic acid	Toluene/water Acetone/water DoOH/CyOH/water	Emulsion	SDS	Extraction	~2.5 *	250 nm	[104]
Amoxicillin	Water/DMSO	Emulsion	SDS	Sensing	45.6	8–10 µm	[97]
Phosphate anion	Water/chloroform	Emulsion	CTAB	Sensing	n.d.	n.d.	[110]
Bovine hemoglobin	Water/toluene	Pickering emulsion	Hb-coated Janus hydroxyapatite NPs	Extraction	4.0	50 µm	[96]
*Listeria Monocytogenes*	Water/DMA	Pickering emulsion	*N*-Acrylchitosan-Quantum Dot	Sensing	4.6	200 µm	[111]
λ-cyhalothrin	Water/hexadecane	Pickering emulsion	Attapulgite particles	Extraction	1.7	50 µm	[112]
Bisphenol A	Water/toluene	Pickering emulsion	Fe_3_O_4_ NPs	Environmental monitoring	1.7	100 µm	[107]
Erythromycin	Water/toluene	Pickering emulsion	Chitosan NPs and Hydrophobic Fe_3_O_4_	Extraction	1.3	53 µm	[113]
Bovine hemoglobin	Water/n-hexane/corn oil	Pickering emulsion	Colloidal casein NPs	Protein Purification	4.1	300 nm	[114]
Gatifloxacin	Cyclohexane/water	Reverse micro-emulsion	Span 60	Extraction	2.0	n.d.	[115]
Kaempferol	ACN/methanol (4:1)	Precipitation	-	Extraction	5.0	8 µm	[103]
Oleanolic acid	Chloroform/methanol (3:1)	Precipitation	-	Extraction	4.8	20 µm	[116]
Huppuric acid	ACN/water	Sol–gel approach	Trifluoro-acetic acid	Extraction	5.1	n.d.	[109]

n.d. = not determined, IFs marked with * were estimated from the rebinding data.

**Table 5 polymers-17-01057-t005:** Summary of MIPs based on ILs as porogens and their IFs and applications.

Template	IL	Co-Solvent	Reaction Type	Application	IF	Ref.
Aesculin	[BMIM]BF_4_	DMSO	Cyclodextrin–bulk polymerization	Drug release	2.4	[129]
Rutin	[BMIM]PF_6_	DMF and ACN	Bulk polymerization	Extraction/Separation	4.8	[130]
Testosterone	[BMIM]BF_4_	Aq. HCl	Condensation/sol–gel approach	Extraction/Separation	13.9	[135]
Chlorogenic Acid	[BMIM]BF_4_	DMSO	Bulk polymerization	Extraction/separation	9.7	[136]
Chicoric Acid	[BMIM]BF_4_	DMSO	Metallic-pivot bulk polymerization	Extraction/Separation	24.8	[137]
Carprofen	[BMIM]BF_4_	DMF and DMSO	RAFT polymerization	Extraction/Separation	1.8	[138]
Norfloxacin	[BMIM]BF_4_	DMF and DMSO	Bulk polymerization	Extraction/Separation	3.4	[139]
Corigalin	[BMIM]BF_4_	DMF and DMSO	Bulk polymerization	Extraction/Separation	9.0	[140]
Isoquercitrin	[BMIM]BF_4_	DMF and DMSO	Bulk polymerization	Extraction/Separation	3.0	[141]
Fluoroquilones	[BMIM]BF_4_	DMSO and CHCl_3_	Molecular Crowding Polymerization	Sensing	3.2	[142]
Dichlorvos	[BMIM]PF_6_	ACN and toluene	Bulk polymerization	Sensing	1.6	[143]
Melamine Triamterene Cyromazine Trimethoprim	[BMIM]BF_4_	MeOH/water	Bulk polymerization	Sensing	2.1–3.9	[144]
Dibutyl Phtalate	[BMIM]BF_4_	CHCl_3_	Bulk polymerization	Sensing	2.0	[145]

## Data Availability

Data sharing is not applicable to this article as no new data were created or analyzed in this study.

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
