# Peer review of "Porogenic Solvents in Molecularly Imprinted Polymer Synthesis: A Comprehensive Review of Current Practices and Emerging Trends"

_polymers, 2025, doi:10.3390/polym17081057_

Round 1

Reviewer 1 Report

Comments and Suggestions for Authors

The paper reviews the role of solvents in the synthesis of molecularly imprinted polymers, especially on the relationship between the solvent and the resulting pore structure and the material performance. The authors review different solvent types (aprotic, protic, IL, DES, supercritical CO2). I think that

  • the topic is most definitely worth the review;
  • the review is very well structured and written
  • it helps navigating the info space (#1 role of review paper as of today) and thus does the job

What is IMHO missing:

  • 1-2 examples (with a figure) connecting the chemistry of the solvent with the resulting materials. The only example as of now is Figure 5, and all schemes related to the synthesis pathways are "not very chemical".
  • The pore structures resulting from solvent use are always described in a vague and qualitative fashion, and I think a more detailed example is worth considering here (including PDS, possible and explain how the solvent chemistry influences that.

This shouldn’t be too difficult to add.

Minor issues:

  • 168 These solvents help ensure that the polymerization focuses on the interactions between monomers and the target molecule, rather than on unintended solvent-polymer interactions. -- “polymerization focuses” is a very strange term
  • However, their environmental impact raises 236 concerns that need to be addressed to balance their use with sustainability efforts [61].
  • "this piece of research"
  • "of template molecules" – the template
  • “chitosan as monomer” -- chitosan is by definition a pseudorandom copolymer
  • 338 A more developed route of generating MIPs that avoids free-radical polymerization and the issues this encounters in aqueous environments is the use of sol-gels -- something wrong with this sentence
  • .[108] --- refs before/after the punctuation, need to be given in the same fashion
  • Fig 5: text in yellow is hardly visible, increase figure size

Comments on the Quality of English Language

English is generally fine, a few examples are given in the comments to the authors

Author Response

What is IMHO missing:

  • 1-2 examples (with a figure) connecting the chemistry of the solvent with the resulting materials. The only example as of now is Figure 5, and all schemes related to the synthesis pathways are "not very chemical".

Thank you for your comment. We have added two figures (4 and 5) with some additional explanations to show the chemical influence of porogen on the sol-gel approach (line 356-370) and on the stability of  the monomer-template complex (line 436-439).

  • The pore structures resulting from solvent use are always described in a vague and qualitative fashion, and I think a more detailed example is worth considering here (including PDS, possible and explain how the solvent chemistry influences that.

We have implemented two examples in the aprotic section (line 225-239) and added some more information for specific examples along the review (lines 213, 444, 604, 691, 733, 779). Unfortunately, most of the publications do not provide chemical explanation on why one porogen outperforms the others.

Minor issues:

  • 168 These solvents help ensure that the polymerization focuses on the interactions between monomers and the target molecule, rather than on unintended solvent-polymer interactions. -- “polymerization focuses” is a very strange term
  • However, their environmental impact raises 236 concerns that need to be addressed to balance their use with sustainability efforts [61].
  • "this piece of research"
  • "of template molecules" – the template
  • “chitosan as monomer” -- chitosan is by definition a pseudorandom copolymer
  • 338 A more developed route of generating MIPs that avoids free-radical polymerization and the issues this encounters in aqueous environments is the use of sol-gels -- something wrong with this sentence
  • .[108] --- refs before/after the punctuation, need to be given in the same fashion
  • Fig 5: text in yellow is hardly visible, increase figure size

Thank you for your comments, we have carefully revised our manuscript.

Reviewer 2 Report

Comments and Suggestions for Authors

This work titled “Porogenic Solvents in Molecularly Imprinted Polymer Synthesis: A Comprehensive Review of Current Practices and Emerging Trends” (Manuscript Number: polymers-3565820) is very well-structured. I agree with its publication after the following major corrections.

Revisions suggestion:

  1. Schematic diagram of the synthesis process of molecularly imprinted polymers should be added.
  2. Table 2 should include performance evaluation indicators for molecularly imprinted polymers, such as adsorption capacity, imprinting factor, etc.
  3. Line 388 mentions the sol-gel method of synthesizing molecularly imprinted polymers, give a schematic of the principle and process of sol-gel synthesis.
  4. Table 3 also should include performance evaluation indicators for molecularly imprinted polymers, such as adsorption capacity, imprinting factor, etc.
  5. How does the use of different types of solvents for the same polymerization method (such sol-gel, bulk etc.) affect MIP?
  6. Table 4 and Table 5 should be elaborated in the main text, for example, what is the selectivity of MIP?

Author Response

Revisions suggestion:

  1. Schematic diagram of the synthesis process of molecularly imprinted polymers should be added.

Thank you for your valuable comments. We have added Figure 1 (line 54) that shows a schematic of the principle behind MIPs and both the chemistry of radical polymerization and sol-gel process.

  1. Table 2 should include performance evaluation indicators for molecularly imprinted polymers, such as adsorption capacity, imprinting factor, etc.

Thank you for your comment, we have added the IF of the mentioned MIPs. We have not included the adsorption capacity since a large number of publications do not include a maximal binding capacity. Similarly for bulk polymerization the particle diameter is often not included.

  1. Line 388 mentions the sol-gel method of synthesizing molecularly imprinted polymers, give a schematic of the principle and process of sol-gel synthesis.

We have added two figures describing the sol-gel process, Figure 1 in the introduction as a comparison to radical polymerization and Figure 4 in the protic section to show a specific example (line 364).

  1. Table 3 also should include performance evaluation indicators for molecularly imprinted polymers, such as adsorption capacity, imprinting factor, etc.

Thank you for your comment, we have added the IF of the mentioned MIPs. We have not included the adsorption capacity since a large number of publications do not include a maximal binding capacity.

  1. How does the use of different types of solvents for the same polymerization method (such sol-gel, bulk etc.) affect MIP?

We have included multiple examples for this trend, for example line 225-239 discussing the impact on the surface area of different solvents. Further solvent impact on polymer affinity are described in line 436-440. MIPs with and without ILs are described in line 547-549. MIPs with and without DES are described in line 603-606. Similarly there are comparisons of MIPs made from traditional solvents with MIPs made from ballmill (line 668-669), supercritical CO2 (line 702-709), MOGs (line 777-783).

  1. Table 4 and Table 5 should be elaborated in the main text, for example, what is the selectivity of MIP?

We have added IF and particle sizes for the synthesized polymers to highlight control over particle size with emulsion and precipitation polymerization.

Round 2

Reviewer 2 Report

Comments and Suggestions for Authors

The manuscript has been well revised and could be accepted.